# Interlaminar Shear Characteristics, Energy Consumption and Carbon Emissions of Polyurethane Mixtures

**Yufeng Bi [1], Min Sun [2,*], Shuo Jing [2], Derui Hou [2], Wei Zhuang [1], Sai Chen [1], Xuwang Jiao [1] and Quanman Zhao [2,*]**

[1]  Shandong Provincial Communications Planning and Design Institute Co., Ltd., Jinan 250031, China; 18866130036@163.com (Y.B.); zhuangweijtky@163.com (W.Z.); chensaiyile@hotmail.com (S.C.); xuwang.jiao@outlook.com (X.J.)

[2]  School of Transportation Engineering, Shandong Jianzhu University, Jinan 250101, China; jingshuo99@163.com (S.J.); hdr09191210@163.com (D.H.)

*  Correspondence: 15253170143@163.com (M.S.); bestcupid@163.com (Q.Z.)

**Abstract:** The interlaminar shear characteristics of a polyurethane (PU) mixture composite structure, quantitatively calculating its energy consumption and carbon emissions were analyzed in this study. Inclined shear tests were carried out on thirteen structures without interlaminar treatment, and high-temperature water bath accelerated loading tests were conducted on three composite structures; further, the interlaminar shear strength of the tire trace position after the accelerated loading test was tested to analyze the influence of both the high-temperature water bath environment and loading on the structure. In addition, based on the medium repair project of the Qingdao-Yinchuan expressway, the construction log of PU mixture pavement and asphalt pavement was investigated. Combined with the calculation parameters provided by the United Nations Intergovernmental Panel on Climate Change (IPCC), the consumption and carbon emissions of the two types of mixtures were calculated and compared quantitatively. The results showed that the shear strength between layers of asphalt mixtures, PU mixture and asphalt mixture was less than 2 MPa; however, the shear strength between PU mixture–cement-stabilized macadam and PU mixture–PU mixture was greater than 3 MPa. Therefore, it was recommended to spread a 0.4 L/m² two-component PU adhesive layer as the interlayer treatment scheme for the structure of asphalt mixture-PU mixture and PU mixture-asphalt mixture; the high-temperature water area and accelerated loading had different effects on the interlaminar shear strength of the three structures. The PUM-16 mixture could effectively reduce energy consumption by 88.3 and 87.2%, carbon emissions by 81.1% and 79.1% in comparison to Stone Matrix asphalt with Nominal maximum particle size of 13.2 mm (SMA-13) and Stone Matrix asphalt with Nominal maximum particle size of 19.0 mm (SMA-20). Thus, the PU mixture was confirmed to be an environmentally friendly road pavement material.

**Keywords:** polyurethane; mixture; shear strength; energy consumption; carbon emission

## 1. Introduction

Asphalt pavement is widely used all over the world; however, its mixing, transportation, paving, and rolling machinery consumes significant amounts of oil and energy, emitting high levels of greenhouse gases such as $CO_2$. Further, asphalt pavement is prone to rutting, looseness, and other issues [1,2]. To solve the aforementioned problems, high molecular polymers polyurethane (PU) is used as binder, the PU mixture pavement is intended to replace asphalt pavement [3]. The entire construction process of PU mixture production, transportation, and compaction is conducted at room temperature, which greatly reduces the consumption of energy consumption and carbon emissions. Further, PU mixtures have better high- and low-temperature stability, fatigue

resistance, and dynamic mechanical properties, which can effectively improve pavement service life and greatly reduce the maintenance frequency and cost [4,5].

Over the past decade, many researchers have conducted studies on composition design, reaction mechanism, and performance characteristics of PU mixtures [6–11]. Wang et al. tested the strength and modulus of Porous Polyurethane Mixture (PPM) by using the unconfined compression test on cement concrete cubes and the flexural test on rectangular beams [6]. Based on the entire load stress test, the strength characteristics and failure mechanism of PPM were studied. Wang et al. prepared a PERS mixture with a porosity of 35% using a two-component PU material provided by BASF Co., Ltd. [7,8]. The invention of a new surface layer material with PU was proposed by Schacht et al., and the vehicle pavement interaction facilities of the Federal Highway Research Institute (BAST) was conducted, it was determined that the surface system has the highest acoustic performance possible. Cong et al. used two-component PU as a binder to replace traditional asphalt binders, and the prepared PPM overcame the limitations of traditional OGFC mixtures [10,11]. However, the curing reaction of PU binders involves a chemical condensation reaction process. Through the reaction of -NCO in the binder with water in the air and active hydrogen on the substrate surface, a cross-linked structure—including a urea bond, amino formate bond, and urea formate macromolecular network—is generated; finally, a three-dimensional network structure is formed, which bonds the substrate together [12].

The strength formation mechanism of PU mixtures differs from that of asphalt mixture and cement concrete. When it is used in conjunction with other road building materials to form a composite pavement structure, a large number of studies on polyurethane mixture composite pavement structure show that the interlayer shear resistance is the key factor affecting its service performance. Therefore, it is necessary to study the interlaminar shear characteristics of composite specimens and further clarify the appropriate interlaminar treatment scheme. Chen et al. carried out interlaminar inclined shear tests to detect the interface shear performance between porous polyurethane mixtures and asphalt sublayers [13]. Zhang et al. studied the effects of waterproof bonding materials, the base surface roughening mode, the upper mixture type, and test conditions on the peak interlaminar shear strength of composite specimens [14]. Mohammad et al. evaluated the effects of waterproof bonding materials, interface types, dry and wet conditions of the base surface, and specimen preparation methods on the peak value of interlaminar shear strength [15]. Luo et al. compared the effects of interface pollution and water saturation conditions on interlayer bonding characteristics [16]. Cao et al. studied the effects of waterproof bonding materials, the base surface roughening mode, the forming mode, the shear mode, and the temperature on the interlayer shear strength [17]. Current research mainly focuses on the influence of different factors (such as the interface roughness mode, waterproof bonding materials, temperature, immersion, freeze–thaw cycles, and interlayer pollution degree) on the interlayer shear strength of asphalt pavement; further, there are some studies on the interlayer shear failure characteristics or mechanism of PU mixture composite pavement structures [18–21].

The greenhouse effect caused by road construction has recently attracted increasing research attention [22]. As a low-carbon and environmentally friendly road building material, the energy conservation and emission reduction effect of PU mixtures in road construction have not been evaluated quantitatively, to the best of our knowledge. Therefore, the energy-saving and emission reduction effect of PU mixture cannot be highlighted, which limits the popularization and application speed of PU mixture pavement. However, the quantitative calculation method for the energy consumption and emissions of road construction based on engineering practice is a relatively mature field of study. Kim et al. established a framework for estimating greenhouse gas emissions [23]. White et al. calculated the average $CO_2$ value of a road by using parameters such as the thickness and characteristics of materials [24]. Cass et al. established a comprehensive life cycle analysis (LCA) model according to an actual project [25]. Wang et al. estimated the

carbon emissions of roads, bridges, and other construction structures in Southwest China [26]. Thives et al. evaluated the carbon dioxide emission and energy consumption generated by road pavement through a literature review [27]. The relationship between energy demand and the environmental impact of pavement in the construction cycle can be analyzed through life cycle analysis (LCA), and the corresponding environmental impact factors have been determined in the past [28–30]. Therefore, these mature methods can be used to quantitatively calculate the energy conservation and emission reduction effect of polyurethane mixture pavement in combination with physical engineering.

In summary, it is necessary to carry out interlaminar shear tests on PU mixture composite specimens, analyze the influence of environmental factors and the load on the interlaminar shear strength, and further clarify the appropriate interlaminar treatment scheme. At the same time, based on the medium repair project of Qingdao–Yinchuan expressway, the construction log of PU mixture pavement and asphalt pavement was investigated. The construction process was divided into three stages: mixture production, mixture transportation, and mixture site construction, which include seven constituent processes. In combination with the calculation parameters provided by the United Nations Intergovernmental Panel on Climate Change (IPCC), a quantitative model of the energy consumption and carbon emission of mixtures was established, which were subsequently compared quantitatively.

## 2. Experiment Scheme

### 2.1. Materials

#### 2.1.1. Raw Material

The materials mainly include a one-component PU binder, two-component PU binder, 70# base asphalt, styrene butadiene styrene (SBS) modified asphalt, and aggregates. The technical indexes of asphalts meet the requirements of JTG F40-2004. One-component PU was used as the binder for the PU mixture, and two-component PU was used as the interlayer bonding material, which are prepared by Wanhua Chemical Co., Ltd. (Yantai, China) [12]. The PU material used is synthesized from 4,4-diphenylmethane diisocyanate (MDI) modified by carbodiimide, polyether polyol, and other additives. It is almost non-toxic and can be used in pavement engineering. The main reaction in the PU synthesis process is the reaction of polyol and diisocyanate to prepare long-chain prepolymer with -NCO end group. The chemical reaction equation is shown in Figure 1. The specific technical indexes of asphalt are given in Table 1 and the indexes of PU binders are given in Table 2.

**Figure 1.** The chemical reaction equation of synthesis process.

**Table 1.** Technical indexes of PU asphalt.

| Technical Indicators | Unit | Technical Requirement | |
|---|---|---|---|
| | | 70# Base Asphalt | SBS Modified Asphalt |
| Penetration | 0.1 mm | 71 | 38 |
| Ductility | cm | >100 (15 °C) | 25.7 (5 °C) |
| Softening point | °C | 48.2 | 81.6 |
| 60 °C dynamic viscosity | Pa.s | 158.7 | 6350.5 |

**Table 2.** Technical indexes of PU binder.

| Technical Indicators | Unit | Technical Requirement | |
|---|---|---|---|
| | | One-Component PU | Two-Component PU |
| Surface drying time | min | 40 ± 10 | 20 ± 5 |
| Tensile strength | MPa | ≥15.0 | ≥20.0 |
| Fracture elongation | % | ≥100 | ≥100 |
| Molecular weight | / | 13,000–17,000 | 15,000–20,000 |

2.1.2. Mixture Material Composition

The technical indexes of the asphalt concrete (AC) and Stone Matrix asphalt (SMA) meet the provisions of JTG F40-2004. The technical indexes of the cement-stabilized macadam mixture meet the provisions of JTJ 034-2018. The Nominal maximum particle sizes of 13.2, 16.0, 19.0, and 26.5 mm are symbolized as 13, 16, 20, and 25, respectively. The mineral aggregate gradation of multi gravel PU concrete (PUM) is designed according to the maximum density theory and the optimum dosage of PU binder is determined by Marshall mix design method. The composition design results of PUM with different aggregate nominal maximum sizes are shown in Table 3.

**Table 3.** Material composition design results of PU mixtures.

| Sieve Size (mm) | Cumulative Passing Percentage of Each Sieve (mm)/% | | |
|---|---|---|---|
| | PUM-13 | PUM-16 | PUM-20 |
| 26.5 | 100 | 100 | 100 |
| 19 | 100 | 100 | 99.7 |
| 16 | 100 | 91.8 | 90.3 |
| 13.2 | 95.6 | 78.4 | 79.4 |
| 9.5 | 66.8 | 60.5 | 63.2 |
| 4.75 | 31.2 | 39.2 | 40.3 |
| 2.36 | 21.5 | 28.4 | 29.5 |
| 1.18 | 16.7 | 19.8 | 21.4 |
| 0.6 | 11.5 | 13.1 | 13.8 |
| 0.3 | 8.6 | 11.3 | 10.4 |
| 0.15 | 6.3 | 7.3 | 7.5 |
| 0.075 | 2.5 | 3.1 | 3.2 |
| Binder content (%) | 5.1 | 5.0 | 5.0 |

2.1.3. Mixture Preparation

The SMA-13, AC-20 and AC-25 mixtures were prepared according to the regulations of "Standard Test Methods of Bitumen and Bituminous Mixtures for Highway Engineering" (E20-2011). The PUM-13, PUM-16, and PUM-20 mixture was prepared using an asphalt mixture mixer; however, the aggregates and binder were at room temperature. First, the specified proportion of aggregates were mixed in the mixing pot for 20–30 s. Next, the corresponding proportion of PU binder was added and mixed for 30–50 s.

Finally, the corresponding proportion of mineral powder was added and mixed for 30–50 s, and the smoothness of the mixture was checked.

*2.2. Inclined Shear Test*

2.2.1. Inclined Shear Test Device

The interlaminar inclined shear test device was used to obtain the interlaminar shear strength of different types of composite specimens. The interlaminar shear test device is designed with reference to the relevant provisions in "Specifications for Design and Construction of Pavement on Highway Steel Deck Bridge" (JTG/T 3364-02-2019) [31]. The inclination angle between the structure layer and the horizontal direction was set to 45°, such that the interlayer had an equivalent shear and normal stress. The controlled strain mode was adopted, with a loading rate of 5 mm/min [10]. The test device used for the interlaminar shear process is shown in Figure 2, and the shear stress corresponding to the peak value of the stress–strain curve was the maximum vertical load. There were five test pieces in each group. The calculation formula for the inclined shear strength is shown in Equation (1), where τ represents the interlaminar shear strength, *F* represents the maximum vertical load, and *S* represents the interface contact area between the two layers.

$$\tau = \frac{F \cdot sin\,45^{\circ}}{S} \tag{1}$$

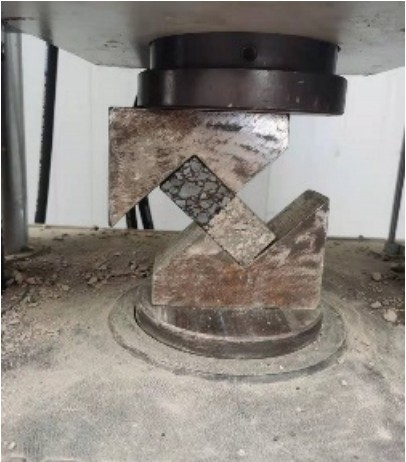

**Figure 2.** Inclined shear test device.

2.2.2. Preparation of the Interlaminar Shear Specimens

A 100 mm thick rutting test mold was used to form the composite rutting plate, and the interlayer shear specimens were then cut and prepared [15]. The specific manufacturing method was as follows: (1) prepare the lower layer mixture rutting test piece with a thickness of 50 mm; (2) install the lower layer test piece onto the rutting test mold with a thickness of 100 mm; (3) carry out the interlayer treatment on the surface of the lower layer according to the treatment scheme; (4) prepare the upper layer mixture and put it into the rutting test mold; (5) form the composite rutting specimen using the wheel rolling method, and remove the formwork after curing; and (6) cut the test piece, remove a 3 cm width from each side (the size of the test pieces after cutting was 50 mm × 50 mm × 100 mm). After the composite specimens were formed, they were cured at room temperature for four days to ensure the complete curing of the PU binder [12]. As far as the PU adhesive layer was concerned, the upper layer mixture was compacted within 0.5–1 h after the two-component PU adhesive layer was spread.

### 2.2.3. Interlaminar Shear Test

Two types of interlaminar shear tests on different composite structures were carried out. Table 4 shows the combination of specimens without interlaminar treatment, whereas Table 5 shows the combination of specimens with interlaminar treatment.

**Table 4.** Composite specimens without adhesive layer.

| Serial Number | I-1 | I-2 | I-3 | I-4 | I-5 | I-6 | I-7 |
|---|---|---|---|---|---|---|---|
| Upper layer | SMA-13 | SMA-13 | PUM-13 | PUM-16 | PUM-16 | PUM-13 | PUM-20 |
| Lower layer | AC-20 | PUM-20 | AC-20 | AC-25 | PUM-16 | PUM-20 | Cement stabilized macadam |

**Table 5.** Composite specimens with adhesive layer.

| Serial Number | II-1 | II-2 | II-3 | II-4 | II-3 | II-5 | II-6 | II-7 | II-6 |
|---|---|---|---|---|---|---|---|---|---|
| Upper layer | | | SMA-13 | | | | PUM-13 | | |
| Adhesive layer | emulsified asphalt | PU | PU | PU | PU+70% macadam (3~5 mm) | PU | PU | PU | PU + 70% macadam (3~5 mm) |
| Content/(L/m²) | 0.5 | 0.2 | 0.4 | 0.6 | 0.4 | 0.2 | 0.4 | 0.6 | 0.4 |
| Lower layer | AC-20 | | PUM-20 | | | | AC-20 | | |

### 2.3. Accelerated Loading Test

In order to clarify the change of interlayer shear strength of composite pavement structure under long-term water temperature and load, the accelerated loading test with high-temperature water area was carried out to simulate the coupling effect of water, temperature and load. On the basis of the interlaminar shear test, structures of I-6, II-1, and II-3 were selected for the accelerated loading test. The forming method of the double-layer rutting plate specimens was the same as that of the inclined shear test [20–22]. The all-environment loading system (ALT-S100) developed by Shandong Jiaotong University was used for the accelerated loading tests (Figure 3a). The test speed was 4.5 km/h, the dynamic load was 1000 kg, the loading speed was 4000 times/h, and the effective working length of the equipment was 1 m. The device shown in Figure 3b was used to arrange the test specimens, where the middle three plates were effective test specimens and the two plates on both sides were cushion blocks.

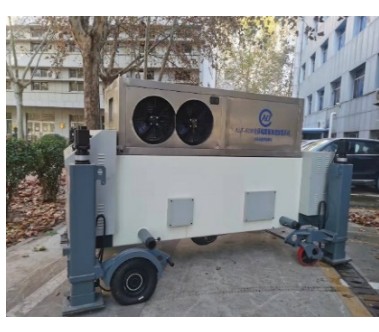 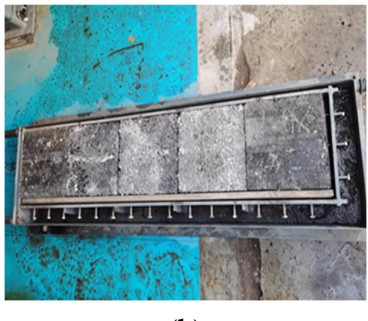 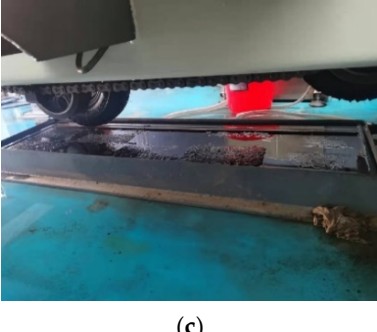

(**a**) (**b**) (**c**)

**Figure 3.** Test equipment and specific working conditions. (**a**) Accelerated loading test device; (**b**) Specimen layout; (**c**) High-temperature water area test.

High-temperature water area loading tests of the specimens with the three structures were carried out, and three plates were formed for each structure. In the high-temperature water area test, 50 °C water was injected into the test tank, the liquid level was flush with

the top of the test piece (Figure 3c), the test specimens were immersed for 8 h in the test environment, and the accelerated loading test was carried out in a circulating water bath environment. The test rolling times were set as 300,000 times.

### 2.4. Energy Consumption and Emission Calculation Method

#### 2.4.1. Project Introduction

Based on the construction log of the medium maintenance project of the Xiajin section of the Qingdao–Yinchuan expressway, data regarding the various equipment and transportation were obtained, and the energy consumption and carbon emission of the mixtures were calculated. The implementation scheme of the project involved removing the floating dust and slag after milling the upper and middle surface of the carriageway, followed by repaving the upper and middle layers. The PU mixture pavement was located in the downward direction of Qingyin–Yinchuan expressway, with a total length of 400 m, from K451 + 600~k451 + 820 to K451 + 840~k452 + 000. The original pavement structure, main line pavement structure, and PU mixture pavement structure are shown in Figure 4.

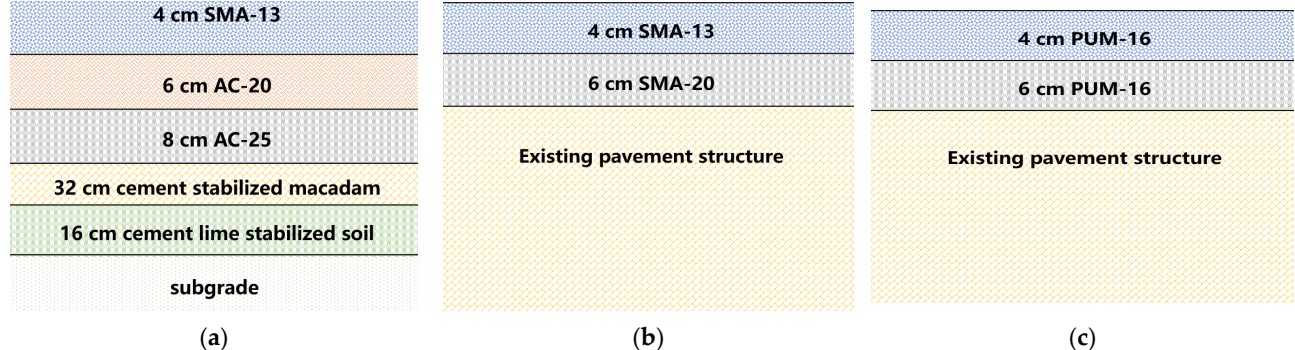

**Figure 4.** Composite pavement structure. (**a**) Old pavement structure; (**b**) Main line pavement structure; (**c**) Polyurethane mixture pavement structure.

#### 2.4.2. Calculation of the Energy Consumption and Carbon Emission during Construction

To calculate the energy consumption and carbon emissions during the mixture construction, the process was divided into three stages: mixture production, mixture transportation, and mixture site construction, which include seven constituent processes. The energy of the mechanical equipment during the construction process was mainly provided by diesel, electricity, and natural gas [27,28]. The carbon emissions generated through the energy consumption and high-temperature volatilization of hot mixtures, the proportions of energy consumption, and the amount of carbon emissions during mixture construction are shown in Figure 5.

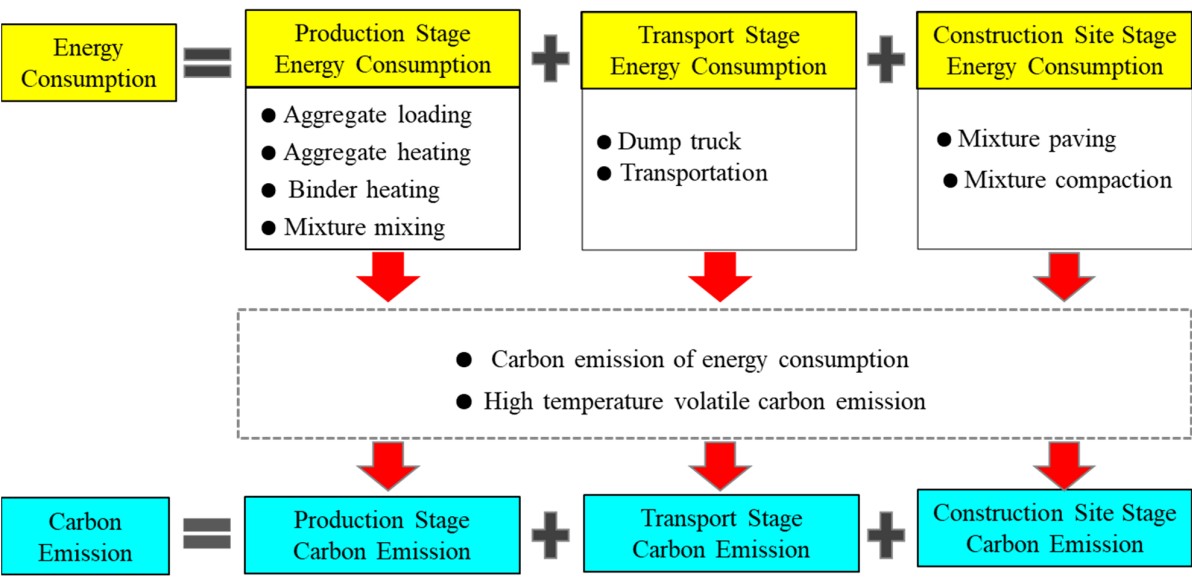

**Figure 5.** Energy consumption and carbon emission of the mixtures.

The carbon emission factor (CEF), energy calorific value (Q), and global warming potential value (GWP) provided by the United Nations IPCC were used as the calculation parameters of the energy consumption and carbon emissions [29,30]. The Q values at various energies are shown in Table 6; the CEF and GWP are shown in Table 7.

**Table 6.** Energy calorific value of energy consumption.

| Energy Type | Unit | Heating Coefficient |
|---|---|---|
| diesel oil | MJ/kg | 42.652 |
| natural gas | MJ/m$^3$ | 38.931 |
| electric power | MJ/(kW·h) | 3.6 |

**Table 7.** CEF and GWP at different energy consumptions.

| | Energy Type | Unit | CO$_2$ | CH$_4$ | N$_2$O |
|---|---|---|---|---|---|
| | diesel oil | Mg/MJ | 74100.0 | 3.0 | 0.6 |
| CEF | natural gas | Mg/MJ | 56100.0 | 1.0 | 0.1 |
| | electric power | Mg/MJ | 317000.0 | / | / |
| | GWP | / | 1 | 28 | 265 |

According to the Q and energy consumption survey data, a quantitative model of the energy consumption during mixture construction was established, as shown in Equation (2). According to the CEF, GWP, and energy consumption survey data, the calculation model of energy consumption and carbon emissions during mixture construction was established [28], as shown in Equations (3)–(5).

$$QC = \sum_{i=1}^{3} \sum_{j=1}^{7} QC_{ij} = \sum_{i=1}^{3} \sum_{j=1}^{7} (M_{ij}Q_{ij} + V_{ij}Q_{ij} + W_{ij}Q_{ij}) \tag{2}$$

$$EC_{ijk} = M_{ij}Q_{ij}C_{ijk} + V_{ij}Q_{ij}C_{ijk} + W_{ij}Q_{ij}C_{ijk} \tag{3}$$

$$EC_{ij} = \sum_{k=1}^{3} (EC_{ijk}p_k)/10^6 \tag{4}$$

$$EC = \sum_{i=1}^{3}\sum_{j=1}^{7} EC_{ij} = \sum_{i=1}^{3}\sum_{j=1}^{7}\sum_{k=1}^{3} \left(EC_{ijk}p_k\right)/10^6 \tag{5}$$

where *QC* is the total energy consumption during pavement construction (MJ/t); EC is the energy consumption emission during pavement construction (kg/t); i represents the three stages of pavement construction, including production, transportation, and construction; j refers to the seven processes during pavement construction, including aggregate supply, aggregate heating, binder heating, mixing, mixture transportation, mixture paving, and mixture compaction; K represents three types of greenhouse gases, including $CO_2$, $CH_4$, and $N_2O$; $EC_{ijk}$ is the emission of K kinds of greenhouse gases corresponding to the jth process of the ith stage (mg/t); $EC_{ij}$ is the energy consumption of carbon emissions corresponding to the jth process of the ith stage (kg/t); $M_{ij}$ is the diesel consumption corresponding to the jth process of the ith stage (kg/t); $V_{ij}$ is the natural gas consumption corresponding to the jth process of the ith stage (m³/t); $W_{ij}$ is the electric energy consumption corresponding to the jth process of the ith stage (kW·h/t); $Q_{ij}$ is the energy calorific value corresponding to the jth process of the ith stage (MJ); $C_{ijk}$ is the energy consumption emission factor of K greenhouse gases corresponding to the jth process of the ith stage (mg/MJ); and $p_k$ is the global warming potential of K greenhouse gases [30].

A large amount of asphalt smoke and dust were produced during the mixing, transportation, paving, and rolling of hot mixtures, which contained gases with significantly high greenhouse effects, including $CO_2$, $CH_4$, and $N_2O$. Therefore, the ZR-3110 multi-gas monitor was used to detect $CO_2$, $CH_4$, and $N_2O$ in asphalt smoke and dust when the pavement was constructed [32,33].

### 2.4.3. Carbon Emission Calculation of the Curing Stage

The strength of the PU mixture was ensured by the curing reaction of the PU binder; water was the key factor affecting the strength of the PU mixture [34]. Further, water in the air could react with the PU binder to form a urea bond, promoting cross-linking and curing.

### Measurement of the NCO Content

The PU binder was dissolved in toluene solution, and the isocyanate in the PU binder reacted with excess di-n-butylamine to generate the corresponding substituted urea. When the reaction was complete, the remaining di-n-butylamine was titrated with a hydrochloric acid standard solution. The reaction equation is shown in Equations (6) and (7).

$$R\text{-NCO} + (C_4H_9)_2NH \;\rightarrow\; RNHCON(C_4H_9)_2 \tag{6}$$

$$(C_4H_9)_2NH + HCL \;\rightarrow\; (C_4H_9)_2NH \cdot HCL \tag{7}$$

The specific steps were as follows:

First, about 4.62 g of the weighed PU binder sample was put into a 250 mL conical flask with a stopper; next, 25 mL of anhydrous toluene was added, the bottle stopper was covered and shaken to completely dissolve the sample [5–8]. Then, 25 mL of 0.1 mol/L di-n-butylamine toluene solution was added with a pipette, the bottle covered with a stopper, shaken for 15 min, and—finally—100 mL of isopropanol and 4–6 drops of bromophenol blue indicator solution were added. The solution was then titrated with 0.1 mol/L hydrochloric acid standard solution. The end point of the reaction was when the solution turned from blue to yellow. The volume of hydrochloric acid used was recorded at this time. Further, blank tests were conducted without adding samples. The mass fraction of isocyanate was calculated according to Equation (8).

$$\omega_{\text{NCO}} = \frac{(V_0 - V_s) \times C \times 42}{1000m} \times 100\% \tag{8}$$

where $\omega_{NCO}$ is the mass fraction of NCO (%); $V_0$ is the volume of hydrochloric acid standard solution consumed by blank titration (mL); vs. is the volume of hydrochloric acid standard solution consumed by sample titration (mL); C is the concentration of the hydrochloric acid solution (mol/L); m is the sampling quantity (g); and 42 is the molar mass of NCO (g/mol).

Calculation of $CO_2$ emissions

After determining the mass fraction of isocyanate, the $CO_2$ emissions per ton of the PU mixture curing reaction can be calculated according to the reaction equation between isocyanate and water (shown in Equation (9)), according to Equation (10):

$$2RNCO + H_2O \rightarrow RNHCONHR + CO_2 \uparrow \tag{9}$$

$$m_{CO_2} = \frac{\omega_{NCO} \times A}{42} \times \frac{1}{2} \times 44 \times 10^3 \tag{10}$$

where $m_{CO_2}$ is the $CO_2$ emission (kg); A is the PU content of the PU mixture (%); and 44 is the molar mass of $CO_2$ (g/mol).

## 3. Results and Discussion

### 3.1. Interlaminar Shear Properties of the Polyurethane Mixture

3.1.1. Interlaminar Shear Properties of Composite Specimens without Interlaminar Treatment

The inclined shear test results of seven composite specimens without interlayer treatment are shown in Figure 6.

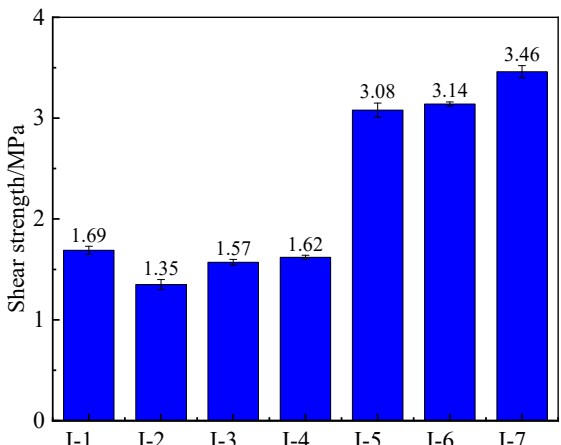

**Figure 6.** Shear strength of specimens without interlaminar treatment.

The data of Figure 6 showed that the shear strengths between layers of asphalt mixture–asphalt mixture and PU mixture–asphalt mixture were less than 2 MPa, and the shear failure mostly occurred at the interlayer. The shear failure form of specimens was shown in Figure 7a–c. The shear strength of I-3 was 0.22 MPa greater than that of I-2, indicating that when the gradations of the upper and lower layers were similar, the interlayer shear strength was higher when the PU mixture used in the upper layer. The shear strength of specimen I-1 was greater than those of I-2, I-3, and I-4, indicating that the interlaminar shear strength between asphalt mixtures was larger than that between PU mixture and asphalt mixture. Therefore, the shear strengths of these structures with different interlaminar treatment schemes are analyzed in the next section.

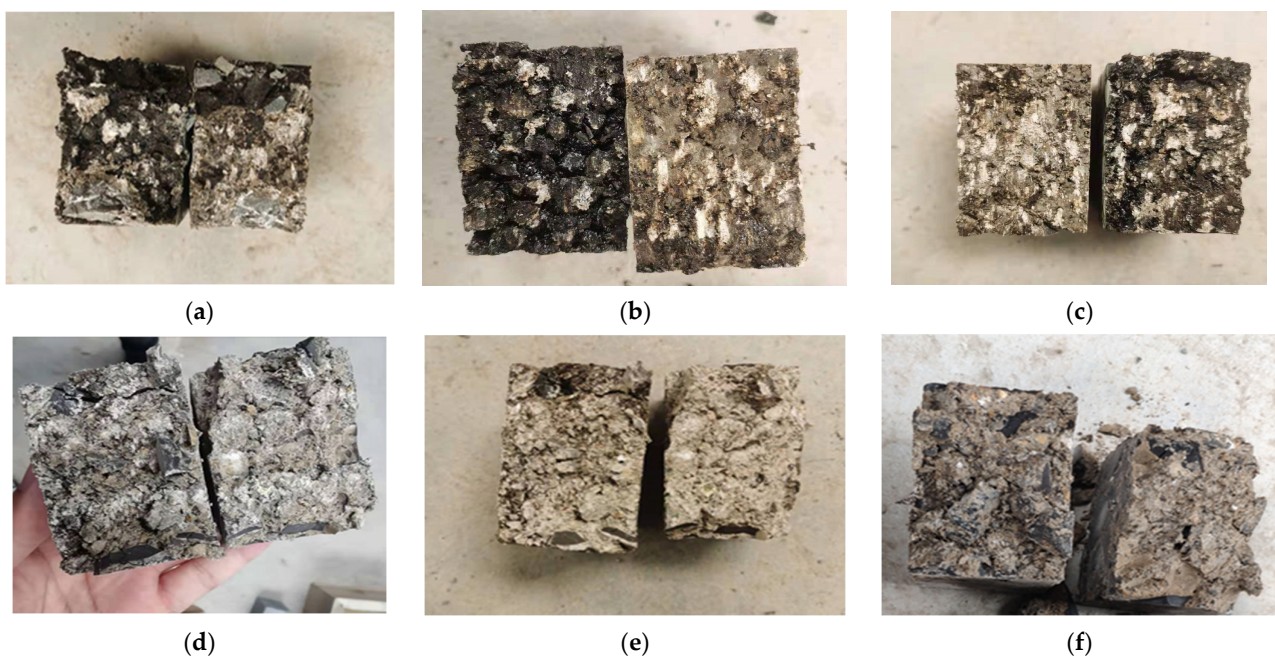

**Figure 7.** Shear failure form of specimens without interlaminar treatment. (**a**) I-1; (**b**) I-2; (**c**) I-3; (**d**) I-5; (**e**) I-6; (**f**) I-7.

The interlayer bonding effect of PU mixtures, PU mixture, and cement-stabilized macadam were better than that of other structures. The interlayer shear strength was greater than 3 MPa, and until shear failure occurred, there was no obvious damage between the layers of these structures, and the interlayer was in a good condition. However, the upper and lower materials were damaged. The shear failure form of specimens without interlaminar treatment was shown in Figure 7d–f. The reason may be that the macromolecular chain of the PU binder between the two layers of PU mixtures produced an interweaving effect, and the macromolecular chain segment of the PU binder between the PU mixture layer and the cement-stabilized macadam layer interacted with the inorganic materials; thus, the interlayer stability was good, and treatment was not required between the layers.

### 3.1.2. Interlaminar Shear Properties of Composite Specimens with Different Treatments

The shear test results for SMA-13 + AC-20, SMA-13 + PUM-20, and PUM-13 + AC-20 with different interlayer treatment schemes are shown in Figure 8.

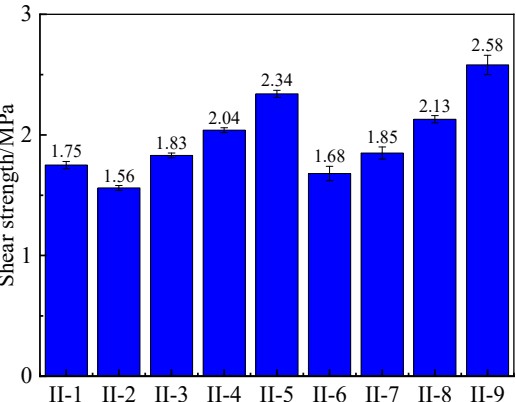

**Figure 8.** Shear strength of specimens.

According to Figure 8, for the SMA-13 + PUM-20 and PUM-13 + AC-20 structures, the interlayer shear strength increased with increases in the distribution amount of the two-component PU adhesive layer. The variation trend of interlayer shear strength with the distribution amount is shown in Figure 9. When the distribution amount of the adhesive layer was 0.4 L/m², the interlayer shear strength of the SMA-13 + PUM-20 structure and PUM-13 + AC-20 structure was greater than that of Ⅱ-1 with the structure of SMA-13 + AC-20 with an adhesive layer. Therefore, when the composite pavement used an SMA-13 + PUM-20 structure and PUM-13 + AC-20 structure, it was recommended to spread 0.4 L/m² two-component PU adhesive layer as the interlayer treatment scheme.

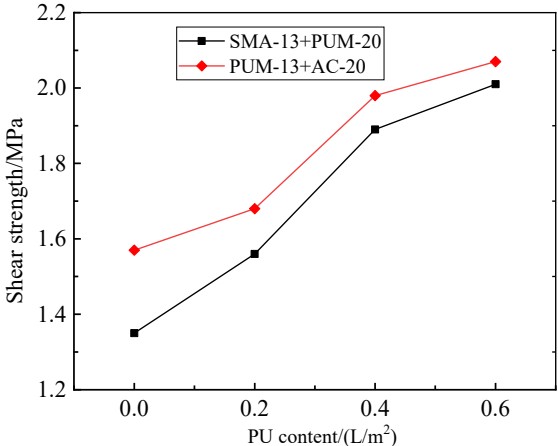

**Figure 9.** Specimens with a PU adhesive layer.

When the two-component PU macadam seal was used in the treatment, the interlayer shear strength was the highest; the interlayer shear strength of the SMA-13 + PUM-20 structure was 2.34 MPa, and the interlayer shear strength of the PUM-13 + AC-20 structure was 2.58 MPa. When the two structures are used for special road conditions, such as heavy loads and long longitudinal slopes, the technology of the two-component PU macadam seal can be used as the interlayer treatment.

*3.2. Long-Term Shear Properties of Composite Specimens*

3.2.1. Surface Properties after Long-Term Accelerated Loading

The tire trace position of the test piece is cut after accelerated loading to a size of 50 mm × 50 mm × 10 mm, and the cut specimens are shown in Figure 10.

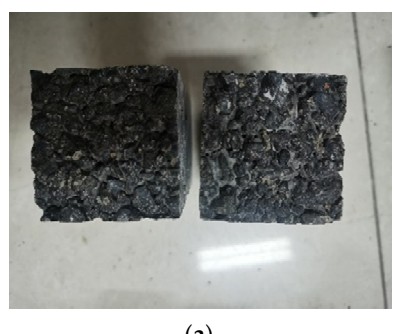 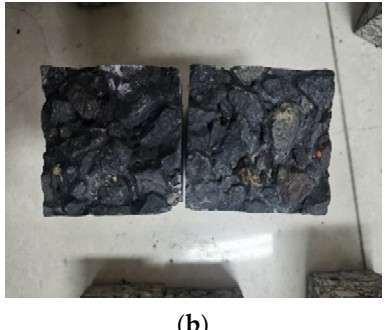 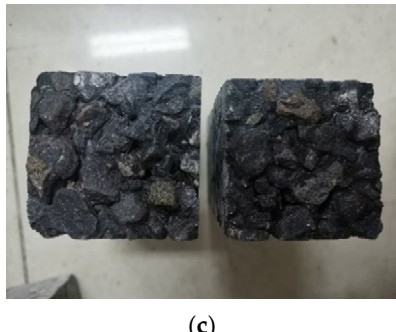

(**a**)　　　　　　　　　　　　　　　(**b**)　　　　　　　　　　　　　　　(**c**)

**Figure 10.** Specimen surfaces after the accelerated loading test. (**a**) I-6; (**b**) II-1; (**c**) II-3.

It can be seen that, after the accelerated loading test, the fine surface aggregate of II-1 and II-3 was mostly missing, while the surface of test piece I-6 basically maintained

integrity, although a small amount of fine aggregate was missing. This shows that, under the dual action of long-term high-temperature water and loading, asphalt mixtures are prone to particle loosening and breaking, while PU mixtures had relatively good stability.

### 3.2.2. Long-Term Interlaminar Shear Property

The shear strength of the three structures before and after accelerated loading tests are shown in Figure 10. The shear strength ratio was calculated by dividing the original shear strength by the shear strength after the accelerated loading test; the results are shown in Figure 11.

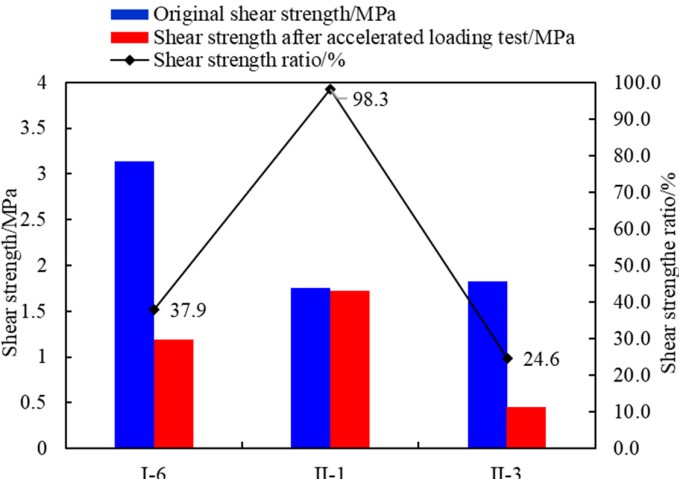

**Figure 11.** Shear strength and shear strength ratio of the three structures.

From Figure 11, it can be seen that the accelerated loading test had different effects on the interlaminar shear strength of the three structures. Among them, the interlaminar shear strength of II-3 decreases the most, and the shear strength ratio after the accelerated loading test was only 24.6%. The specimen after the shear test is shown in Figure 12c, and the failure surface between the two layers was found to be very smooth. This indicates that the high-temperature water area and loading greatly reduced the interlayer shear strength of the composite structure of the PU mixture–asphalt mixture; therefore, this structure was not suitable for high-temperature and wet environments.

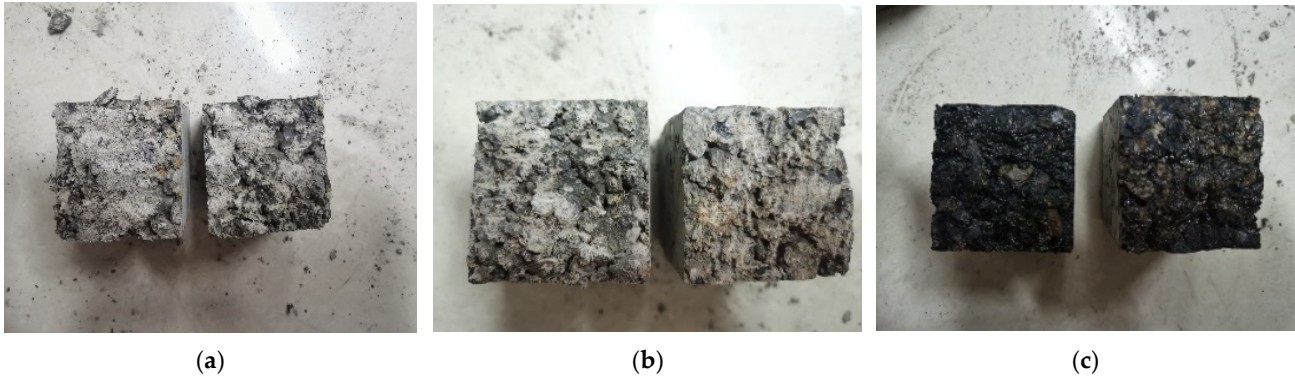

**Figure 12.** Failure form of the shear test after accelerated loading test. (**a**) I-6; (**b**) II-1; (**c**) II-3.

The interlaminar shear strength of the double PU mixture layer structure (I-6) was greater than 3 MPa; however, the shear strength ratio after the accelerated loading test was only 37.9%. The sheared specimen is shown in Figure 12a, indicating that the high-

temperature water area and load have a great impact on the interlaminar shear strength of the structure. A large number of studies show that when the interlaminar shear strength of pavement structure is greater than 1MPa, it can meet the basic service requirements of pavement [28]. The interlaminar shear strength after accelerated loading was greater than 1 MPa, thus it can meet the basic requirements of pavement structures.

The interlaminar shear strength of the double asphalt mixture layer structure specimen (II-1) changed slightly before and after accelerated loading, and the shear strength ratio after accelerated loading was 98.3%. The sheared specimen is shown in Figure 12b, indicating that the high-temperature water area and loading have little impact on the interlaminar shear characteristics of the double-layer asphalt mixture structure.

### 3.3. Evaluation of the Energy-Saving and Emission-Reduction Effects of Composite Pavement

3.3.1. Energy Consumption during Pavement Construction

According to the site construction log, the data of various equipment were obtained for energy consumption calculation. The energy consumption of each construction stage and the process of the three mixtures are shown in Table 8.

**Table 8.** Energy consumption list of mixtures at each stage.

| Material Type | Production Stage | | | | Transportation Stage | Construction Site Stage | |
| --- | --- | --- | --- | --- | --- | --- | --- |
| | Aggregate Feed (Diesel)/(L·t⁻¹) | Aggregate Heating (Natural Gas)/(m³·t⁻¹) | Binder Heating (Natural Gas)/(m³·t⁻¹) | Mixture Mixing (Electric Energy)/(kW·h) | Mixture Transportation (Diesel)/(L·t⁻¹) | Mixture Paving (Diesel)/(L·t⁻¹) | Mixture Rolling (Diesel)/(L·t⁻¹) |
| SMA-13 | 0.195 | 8148 | 1353 | 3200 | 0.27 | 0.182 | 0.277 |
| SMA-20 | 0.142 | 7276 | 1279 | 3011 | 0.43 | 0.155 | 0.231 |
| PUM-16 | 0.158 | 0 | 0 | 3342 | 0.36 | 0.167 | 0.188 |

The energy consumption and carbon emissions during the construction process per ton of the mixture were calculated using Equations (2)–(5). The results are shown in Figure 13, and the proportions of energy consumption of different construction stages of the three mixtures are shown in Figure 14.

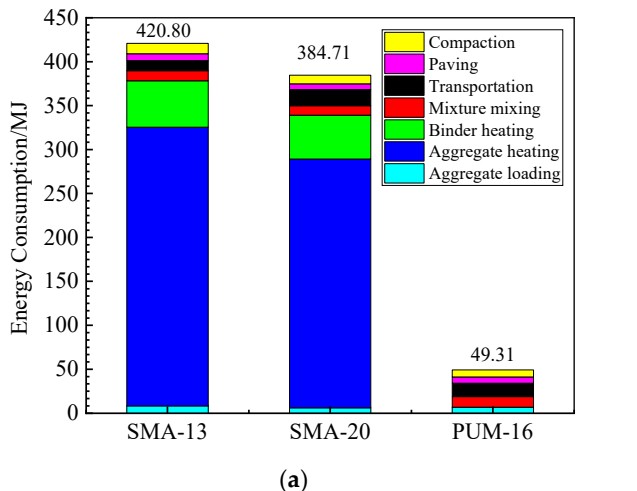
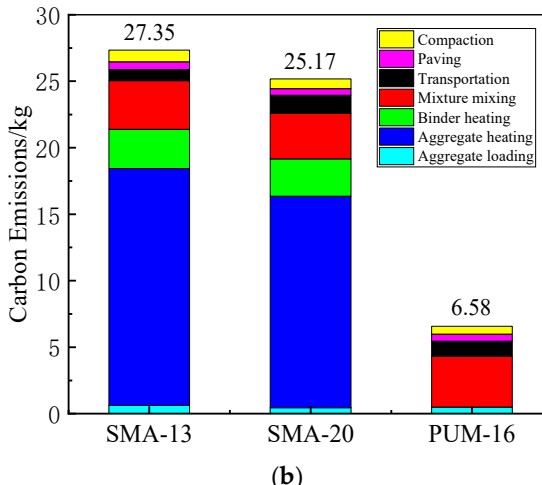

(**a**)　　　　　　　　　　　　　　(**b**)

**Figure 13.** Energy consumption and carbon emissions of pavement construction. (**a**) Energy consumption; (**b**) Carbon emissions.

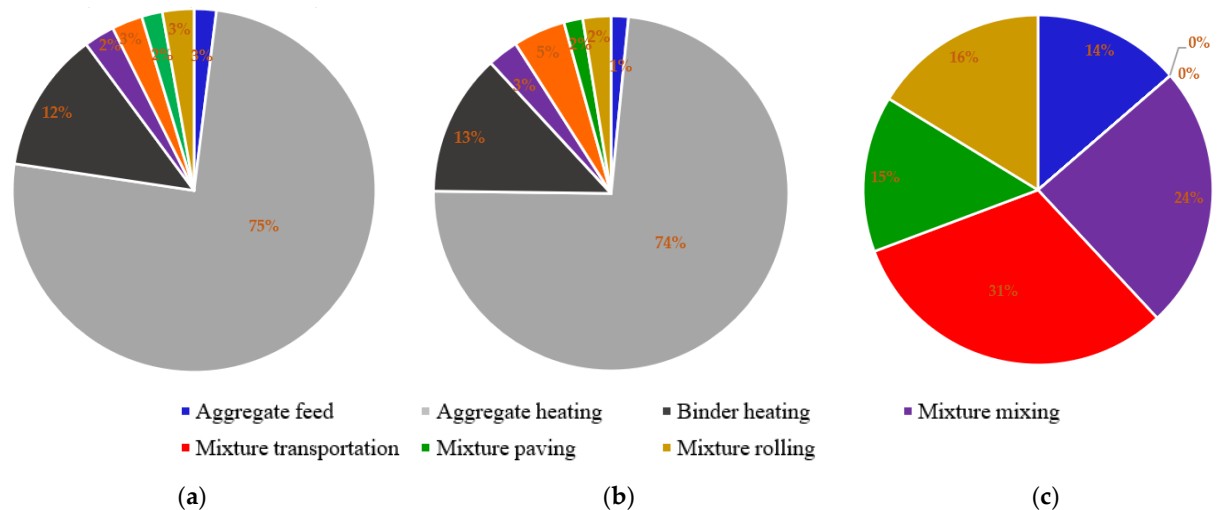

**Figure 14.** Energy consumption of pavement construction. (**a**) SMA-13; (**b**) SMA-20; (**c**) PUM-16.

According to Figures 13 and 14, the energy consumption of the PUM-16 mixture in the construction stage was 11.7% and 12.8% of that of SMA-13 and SMA-20, respectively, and the carbon emissions were 24.2% and 26.1% of that of SMA-13 and SMA-20, respectively, indicating that the PUM-16 mixture could effectively reduce the energy consumption and carbon emissions of the construction process and meet the requirements of energy conservation and emission reduction.

For the asphalt mixture, the energy consumption and emission for aggregate heating accounted for the largest proportion of the whole, followed by asphalt heating and asphalt mixing. The energy consumption for aggregate heating, asphalt heating, and mixture mixing accounted for about 75%, 12%, and 3% of the total energy consumption, respectively. For the PU mixture, the energy consumption of mixture transportation and mixing was the largest, accounting for about 26.1% and 24% of the entire construction process, respectively, followed by that for compaction, paving, and aggregate feed.

### 3.3.2. Carbon Emission during the High-Temperature Volatilization and Curing Stage

The high-temperature volatile carbon emissions of the SMA-13 and SMA-20 mixtures during each construction process were detected. Owing to the normal temperature construction of the PUM-13 mixture, there were no high-temperature volatile carbon emissions. The results are shown in Table 9.

**Table 9.** High-temperature volatile carbon emissions from hot-mix asphalt mixture kg·t$^{-1}$.

| Mixture Type | Mixing/kg·t$^{-1}$ | Transportation/kg·t$^{-1}$ | Paving/kg·t$^{-1}$ | Compaction/kg·t$^{-1}$ | Sum/kg·t$^{-1}$ |
|---|---|---|---|---|---|
| SMA-13 | 0.040 | 0.247 | 1633 | 19.468 | 21.388 |
| SMA-20 | 0.035 | 0.1888 | 1509 | 17.085 | 18.818 |
| PUM-16 | 0 | 0 | 0 | 0 | 0 |

The NCO content of the PU mixture was calculated as 10%, and the $CO_2$ produced during the curing reaction of 1t of the PUM-13 mixture was 2.62 kg.

### 3.3.3. Total Energy Consumption and Carbon Emissions of the Three Mixtures

The carbon emissions of the three mixtures during the construction stage, high-temperature volatilization stage, and curing stage can be summed, as shown in Figure 15. It can be seen that the PUM-16 mixture can effectively reduce the energy consumption by 88.3% and 87.2% and carbon emissions by 81.1% and 79.1%, respectively, in comparison to SMA-13 and SMA-20. From the perspective of the medium maintenance project of

Qingdao–Yinchuan expressway, the pavement structure of PUM-16 + PUM-16 was compared with the structure of SMA-20 + SMA-13; it was found that the energy consumption was reduced by 87.8%, the carbon emissions were reduced by 80.2%, and the energy conservation and emission reduction effects of the PUM-16 + PUM-16 structure were remarkable.

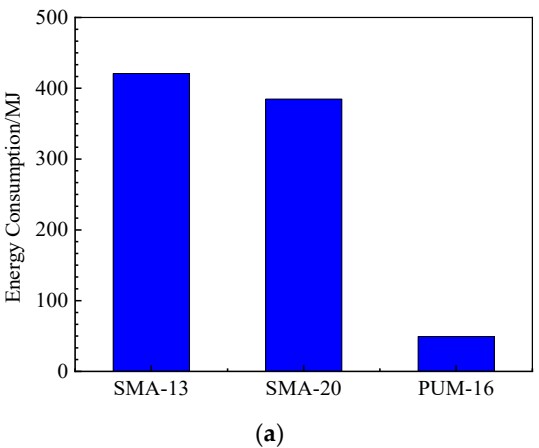
(**a**)

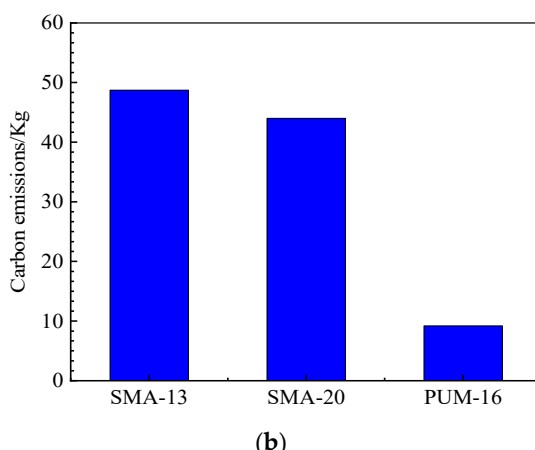
(**b**)

**Figure 15.** Energy consumption and carbon emissions of the three mixtures. (**a**) Energy consumption; (**b**) Carbon emissions.

## 4. Conclusions

The following conclusions can be drawn:

1. The shear strength between the layers of asphalt mixtures, PU mixture, and asphalt mixture was less than that of the PU mixture–cement-stabilized macadam and PU mixture–PU mixture structures.
2. It is recommended to spread 0.4 L/m² of two-component PU adhesive layer as the interlayer treatment scheme for the SMA-13 + PUM-20 structure and PUM-13 + AC-20 structure.
3. The long-term coupling action of water, temperature, and load has different effects on different composite pavement structures. Under the coupling action, the shear strength of the double PU mixture layer structure, the PU mixture, and asphalt mixture composite structure decreased greatly, while the shear strength of the double asphalt mixture layer decreased slightly.
4. In comparison to asphalt mixtures, the energy conservation and emission reduction effect of the PU mixture was remarkable.

**Author Contributions:** Conceptualization, Y.B.; Methodology, S.C.; Validation, M.S. and S.J.; Formal analysis, M.S. and Q.Z.; Investigation, Y.B.; Resources, W.Z. and X.J.; Data curation, S.J. and D.H.; Writing—original draft preparation, M.S. and X.J.; Writing—review and editing, W.Z.; Supervision, S.C. and D.H.; Project administration, Q.Z. and W.Z. All authors have read and agreed to the published version of the manuscript.

**Funding:** This work was financially supported by Key Scientific Research Projects in the Transportation Industry of the Ministry of Transport (2019MS2028), Shandong Expressway Group Project (HSB 2021-72).

**Institutional Review Board Statement:** Not applicable.

**Informed Consent Statement:** Not applicable.

**Data Availability Statement:** This original copy does not include distributed figures, tables, and charts before, thus all figures, tables, and charts of this original copy are unique.

**Acknowledgments:** We would like to recognize numerous co-workers, students, and research facility associates for giving specialized assistance on instrument examination.

**Conflicts of Interest:** The authors declare no conflicts of interest.

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
