# Peer review of "Interlaminar Shear Characteristics, Energy Consumption, and Carbon Emissions of Polyurethane Mixtures"

_coatings, doi:10.3390/coatings12030400_

Round 1

Reviewer 1 Report

The article makes a very good impression and seems to be prepared at a high level.
A detailed literature review is made, the raw materials used and the applied methods and algorithms are described in detail. The studies are described in detail; the results obtained are given a detailed and logical description.
However, there are minor remarks that are recommended to be taken into account to improve the quality of the article:

  1.  In the Introduction section, a very good literature review is made on the chosen topic, but it is not clear what the scientific novelty and the main purpose of the study are. It is recommended that these elements be highlighted separately at the end of the Introduction section.
  2.  What do the numbers in the following abbreviations symbolize: SMA-13, 16, 20; PUM-13, 16, 20; AC-20, 25?

Author Response

The article makes a very good impression and seems to be prepared at a high level. A detailed literature review is made, the raw materials used and the applied methods and algorithms are described in detail. The studies are described in detail; the results obtained are given a detailed and logical description. However, there are minor remarks that are recommended to be taken into account to improve the quality of the article:

Comment 1: In the Introduction section, a very good literature review is made on the chosen topic, but it is not clear what the scientific novelty and the main purpose of the study are. It is recommended that these elements be highlighted separately at the end of the Introduction section.

Response: Thanks to the reviewer for affirmation of the research direction of the article. The authors have cleared the relationship of the scientific novelty and the main purpose of the study.

Therefore, the energy-saving and emission reduction effect of PU mixture cannot be highlighted, which limits the popularization and application speed of PU mixture pavement.

Therefore, these mature methods can be used to quantitatively calculate the energy conservation and emission reduction effect of polyurethane mixture pavement in com-bination with physical engineering.

See line 69-71, line 92-94 and line 107-109.

Comment 2: What do the numbers in the following abbreviations symbolize: SMA-13, 16, 20; PUM-13, 16, 20; AC-20, 25?

Response: Thank you very much. The numbers in the following abbreviations symbolize of SMA-13, 16, 20; PUM-13, 16, 20; AC-20, 25 are the Nominal maximum particle size of the mixture. The Nominal maximum particle sizes of 13.2 mm, 16.0 mm, 19.0 mm, 26.5 mm are symbolized as 13, 16, 20, 25 respectively. The authors have cleared the definition.

See Page 4, line 146-147.

Reviewer 2 Report

In this paper the interlaminar shear characteristics of a polyurethane (PU) mixture composite structure was investigated based on calculating of its energy consumption and carbon emissions. The paper has some valuable results but there are some points and comments that are need to be carefully considered by the authors before publication as they are coming in the following paragraphs:

1-Page 3, Table 1, How was the optimum binder content determined for different PUM mixtures? The authors need to explain about that.

2-Page 4, line 150: “for 4 d” has to be replaced by “for 4 days”.

3-Page 10, Fig.8: It seems that in the figure legend, SPU is wrong and should be changed by PUM.

4-Page 10, line 302-304: it has been stated that “In Figure 8 when the distribution amount of the adhesive layer was 0.4 L/m2, the interlayer shear strength of the SMA-13+PUM-20 structure and PUM-13+AC-20 structure was greater than that of the SMA-13 + AC-20 structure with an adhesive layer”. But there is no SMA-13 + AC-20 structure in Fig.8.

5-Page 11, Section 3.2.1: in this section the authors need to explain the reason for doing the accelerated loading test. What was the aim and the purpose of doing this costly and time consuming test?

6-Page 12, line 343-344: It was stated that “The interlaminar shear strength after accelerated loading was greater than 1 MPa, which can still meet the requirements of pavement structures”. It need to be determined that according to which standard the shear strength of 1 MPa meet the requirement of pavement structure.

7-Page 15, conclusion, no.3, line 406-409. It is not clear what the meaning of this part is. Does it mean that the results obtained from accelerated loading test is contradicted with what concluded from other test? It need to be explained more clearly and clarify that.

8-The authors need to strengthen the literature review by adding more relevant papers including the followings:

- Investigate the use of recycled polyvinyl chloride (PVC) particles in improving the mechanical properties of stone mastic asphalt (SMA), Construction and Building Materials, 2022.

- The effects of nano zinc oxide (ZnO) and nano reduced graphene oxide (RGO) on moisture susceptibility property of stone mastic asphalt (SMA), Case Studies in Construction Materials, 2021.

Author Response

In this paper the interlaminar shear characteristics of a polyurethane (PU) mixture composite structure was investigated based on calculating of its energy consumption and carbon emissions. The paper has some valuable results but there are some points and comments that are need to be carefully considered by the authors before publication as they are coming in the following paragraphs:

Comment 1: Page 3, Table 1, How was the optimum binder content determined for different PUM mixtures? The authors need to explain about that.

Response: Thank you very much. The mineral aggregate gradation of multi gravel PU concrete (PUM) was designed according to the maximum density theory and the optimum dosage of PU binder was determined by Marshall mix design method. The authors have explained the composition design method of PUM.  See Page 4, line 148-151.

Comment 2: Page 4, line 150: “for 4 d” has to be replaced by “for 4 days”.

Response: Thank you very much.  The authors have revised the mistake. See Page 5, line 194.

Comment 3: Page 10, Fig.8: It seems that in the figure legend, SPU is wrong and should be changed by PUM.

Response: Thank you very much. The authors have revised Figure 8. See Page 11, line 348.

Comment 4: Page 10, line 302-304: it has been stated that “In Figure 8 when the distribution amount of the adhesive layer was 0.4 L/m2, the interlayer shear strength of the SMA-13+PUM-20 structure and PUM-13+AC-20 structure was greater than that of the SMA-13 + AC-20 structure with an adhesive layer”. But there is no SMA-13 + AC-20 structure in Fig.8.

Response: Thank you very much. The SMA-13 + AC-20 structure with an adhesive layer is defined as â…¡-1. And the interlayer shear strength of the â…¡-1 was 1.75 MPa, the interlayer shear strength of the SMA-13+PUM-20 structure and PUM-13+AC-20 structure was greater than 1.75 MPa. The statement in the previous manuscript is not clear, and the author has made corresponding modifications.  See Page 11, line 355.

Comment 5: 5-Page 11, Section 3.2.1: in this section the authors need to explain the reason for doing the accelerated loading test. What was the aim and the purpose of doing this costly and time consuming test?

Response: Thank you very much.  In order to clarify the change of interlayer shear strength of composite pavement structure under long-term temperature and load, the accelerated loading test with high-temperature water area was carried out to simulate the coupling effect of water, temperature and load. The authors have explained the reason for doing the accelerated loading test. See Page 6, line 205-208.

Comment 6: Page 12, line 343-344: It was stated that “The interlaminar shear strength after accelerated loading was greater than 1 MPa, which can still meet the requirements of pavement structures”. It needs to be determined that according to which standard the shear strength of 1 MPa meet the requirement of pavement structure.

Response: Thank you very much. A large number of studies show that when the interlaminar shear strength of pavement structure is greater than 1MPa, it can meet the basic service requirements of pavement. And the interlaminar shear strength after accelerated loading was greater than 1 MPa, thus it can meet the basic requirements of pavement structures.

See Page 13, line 397-400.

Comment 7: Page 15, conclusion, no.3, line 406-409. It is not clear what the meaning of this part is. Does it mean that the results obtained from accelerated loading test is contradicted with what concluded from other test? It need to be explained more clearly and clarify that.

Response: Thank you very much.  The description of this part is not clear. The authors have revised in the manuscript. The long-term coupling action of water, temperature and load has different effects on different composite pavement structures. Under the coupling action, the shear strength of the double PU mixture layer structure, the PU mixture and asphalt mixture composite structure decreased greatly, while the shear strength of the double asphalt mixture layer decreased slightly.

See Page 16, line 468-472.

Comment 8: The authors need to strengthen the literature review by adding more relevant papers including the followings:

- Investigate the use of recycled polyvinyl chloride (PVC) particles in improving the mechanical properties of stone mastic asphalt (SMA), Construction and Building Materials, 2022.

- The effects of nano zinc oxide (ZnO) and nano reduced graphene oxide (RGO) on moisture susceptibility property of stone mastic asphalt (SMA), Case Studies in Construction Materials, 2021.

Response: Thank you very much.   The authors have added the relevant papers in the manuscript.

Reviewer 3 Report

The work is interesting but it looks to me that the background study was not investigated well. There are many studies in this area. In fact, the design of pavement using mechanistic method is the key research area at present. This study has some new finding. But it is impossible to understand the motivation and improvement with comparison to the existing knowledge. Please discuss the literature more elaborately using more examples.

Most, if not all, of the references used are local to authors. Please use some international references. One example, pavement design – materials, analysis and highways, mcgraw hill.  

The conclusion section can be rewritten to concisely give the conclusion. In current state, this section became a discussion section. Or, a new conclusion section can be added.

Author Response

Comment 1: The work is interesting but it looks to me that the background study was not investigated well. There are many studies in this area. In fact, the design of pavement using mechanistic method is the key research area at present. This study has some new finding. But it is impossible to understand the motivation and improvement with comparison to the existing knowledge. Please discuss the literature more elaborately using more examples.

Response: Thank you very much. The authors have added and investigated more studies in the introduction section. The motivation and improvement with comparison to the existing knowledge was descripted. And more examples were used to elaborate the literature. See Page 2, Line 69-71, 92-94, Page 3, Line 107-109.

Comment 2: Most, if not all, of the references used are local to authors. Please use some international references. One example, pavement design – materials, analysis and highways, mcgraw hill.  

Response: Thank you very much. The authors have revised and used more international references. Some references of pavement design, materials, analysis and highways were added and analyzed systemically. See Page 17, line 564-569.

Comment 3: The conclusion section can be rewritten to concisely give the conclusion. In current state, this section became a discussion section. Or, a new conclusion section can be added.

Response: Thank you very much. The conclusion section was rewritten by the authors. The conclusions were given concisely. See Page 15, Line 457-470.

Reviewer 4 Report

A thorough grammar check is important. Just after Abstract "The" is bold needs to be fixed. The abstract needs to be revised and shortened (280+ words). Please explain if you want to use any abbreviated words.

Some other comments:
1. PU is harmful, a section must be added on the toxicity of the PU before even it's application. 

2. Before resubmitting please fully explain all abbreviations. Some are:

Page 2, line 47: PPM?

Page 2, line 51: BASF?

Please give a thorough revision for this kind of confusion. 

3. Please add technical indexes of asphalt binders and also generic properties for the one and two-component PUs

4. Please add details of asphalt concrete preparation as well as macadam mixture.

5. Is there any standards for inclines shear test? What is the justification for using it?

6. Page 4, line 149: How practical is 4 days curing for PU binder in comparison with a conventional binder.

7. All the figures should have replicates and standard deviations. A statistical significance test is necessary. 

8. Please embed the %percentages in Figure 13.

Author Response

A thorough grammar check is important. Just after Abstract "The" is bold needs to be fixed. The abstract needs to be revised and shortened (280+ words). Please explain if you want to use any abbreviated words.

Response: Thank you very much. The authors have checked the grammar thoroughly. The abstract was revised and shortened. The words number of the abstract is 287 of this manuscript. The abbreviated words used in the manuscript were explained carefully. And “The” was revised too. See Page 1, Line 11-31.

Comment 1: PU is harmful, a section must be added on the toxicity of the PU before even it's application. 

Response: The PU material used is synthesized from 4,4-dipheylmethane diisocyanate (MDI) modified by carbodiimide, polyether polyol and other additives. It is almost non-toxic and can be used in pavement engineering. See Page 3, Line 129-131.

Comment 2: Before resubmitting please fully explain all abbreviations. Some are:

Page 2, line 47: PPM?

Page 2, line 51: BASF? BASF Co., Ltd

Please give a thorough revision for this kind of confusion. 

Response: Thank you very much. The authors have fully explained all abbreviations. The PPM is the abbreviation of Porous Polyurethane Mixture. The BASF Co., Ltd is the PU manufacturer company name. The authors have thoroughly revised the description.

See Page 2, Line 52, 56.

Comment 3: Please add technical indexes of asphalt binders and also generic properties for the one and two-component Pus.

Response: Thank you very much. The authors have added the technical indexes of asphalt binders and also generic properties for the one and two-component Pu. See Table 1 and Table 2. See Page 3, Line 133, 134.

Comment 4: Please add details of asphalt concrete preparation as well as macadam mixture.

Response: Thank you very much. The authors have added details of asphalt concrete preparation as well as macadam PU mixture. A new chapter of Mixture preparation was added. See Page 4, Line 151-159.

Comment 5: Is there any standards for inclines shear test? What is the justification for using it?

Response: Thank you very much. The interlaminar shear test device is designed with reference to the relevant provisions in “Specifications for Design and Construction of Pavement on Highway Steel Deck Bridge” (JTG/T 3364-02-2019). The inclination angle between the structure layer and the horizontal direction was set to 45°, such that the interlayer had an equivalent shear and normal stress. See Page 4, Line 163-165.

Comment 6: Page 4, line 149: How practical is 4 days curing for PU binder in comparison with a conventional binder.

Response: Thank you very much. The authors have previously carried out research on the strength formation mechanism of PU mixture. After 24 hours of curing, the splitting strength of PU mixture is greater than 1MPa, so the next layer can be constructed or the traffic can be opened. With the increase of curing time, the strength of polyurethane mixture gradually increases, and the strength basically forms after 4 days of curing. Thus, PU mixture is suitable for the conventional asphalt construction work. These findings are described in detail in reference 12. The reference 12 is mentioned in the part. See Page 5, Line 192.

Comment 7: All the figures should have replicates and standard deviations. A statistical significance test is necessary. 

Response: Thank you very much. The authors have conducted statistical analysis and added the standard deviations in some Figures. See Page 9, Line 321 and Page 11, Line 355.

Comment 8: Please embed the %percentages in Figure 13.

Response: Thank you very much.  The authors have embedded the percentage in Figure 13. See Figure 13. See Page 14, Line 418.

Round 2

Reviewer 3 Report

thanks for revising the article

Author Response

Thank the reviewer for your approval of the manuscript.

Reviewer 4 Report

The authors have addressed most of my comments. They could expand a little more on synthesis of this PU, not by just providing only single line synthesis process. 

Author Response

The authors have addressed most of my comments. They could expand a little more on synthesis of this PU, not by just providing only single line synthesis process. 

Response: Thank you very much for your comments. The authors have revised the manuscript. The main reaction in the PU synthesis process is the reaction of polyol and diisocyanate to prepare long-chain prepolymer with -NCO end group. The chemical reaction equation is shown in Figure 1.  See Page 3 Line 131-133 and Figure 1.